# Comparative Study of the Lipophilicity of Selected Anti-Androgenic and Blood Uric Acid Lowering Compounds

**DOI:** 10.3390/molecules28010166

**Published:** 2022-12-25

**Authors:** Dawid Wardecki, Małgorzata Dołowy, Katarzyna Bober-Majnusz, Josef Jampilek

**Affiliations:** 1Doctoral School of the Medical University of Silesia in Katowice, Faculty of Pharmaceutical Sciences in Sosnowiec, 41-200 Sosnowiec, Poland; 2Department of Analytical Chemistry, Faculty of Pharmaceutical Sciences in Sosnowiec, Medical University of Silesia in Katowice, Jagiellońska 4, 41-200 Sosnowiec, Poland; 3Department of Analytical Chemistry, Faculty of Natural Sciences, Comenius University, Ilkovicova 6, 842 15 Bratislava, Slovakia

**Keywords:** heterocycles, lipophilicity, TLC, anti-androgens, blood uric acid lowering compounds, principal component analysis, cluster analysis, SRD analysis

## Abstract

This study aimed to evaluate the lipophilicity of a series substances lowering the concentration of uric acid in blood and anti-androgen drugs by thin-layer chromatography in reversed-phase systems (RP-TLC, RP-HPTLC) and computational methods. The chromatographic parameter of lipophilicity (R_MW_) of tested compounds was determined on three stationary phases, i.e., RP18F_254_, RP18WF_254_ and RP2F_254_, using ethanol–water, propan-2-ol-water and acetonitrile–water in various volume compositions as mobile phases. The chromatographic analysis led to determining the experimental value of the lipophilicity parameter for each of the tested compounds, including those for which the experimental value of the partition coefficient (logP_exp_) as a measure of lipophilicity is not well described in available databases, such as febuxostat, oxypurinol, ailanthone, abiraterone and teriflunomide. The chromatographic parameters of lipophilicity were compared with the logP values obtained with various software packages, such as AClogP, AlogPs, AlogP, MlogP, XlogP2, XlogP3, ACD/logP and logP_KOWWIN_. The obtained results indicate that, among selected chromatographic parameters of lipophilicity, both experimental and calculated logP values gave similar results, and these RP-TLC or RP-HPTLC systems can be successfully applied to estimate the lipophilicity of studied heterocyclic compounds belonging to two different pharmacological groups. This work also illustrates the similarity and difference existing between the tested compounds under study using the chemometric methods, such as principal component analysis (PCA) and cluster analysis (CA). In addition, a relatively new approach based on the sum of ranking differences (SRD) was used to compare the chromatographically obtained and theoretical lipophilicity descriptors of studied compounds.

## 1. Introduction

Among the various properties, the most important is lipophilicity, which is involved in the absorption, distribution, metabolism, elimination and toxicity (i.e., ADMET) of bioactive substances [1]. Lipophilicity is successfully used in the drug discovery process in both QSAR (quantitative structure–activity relationship) and also QSPR (quantitative structure–property relationship) studies of new drug candidates [2,3]. Therefore, numerous pieces of work are currently underway to evaluate the lipophilicity parameters of the different classes of compounds, including heterocycles [4,5,6,7,8,9,10,11,12,13,14,15,16,17,18,19,20,21]. The lipophilicity of a molecule is usually represented by partition coefficient (P) or by logP determined by an experimental method, such as shake flask in n-octanol system or in silico using different algorithms based on the structure of the appropriate molecule [1]. Recently, many commercial and free software packages are available for such calculations, e.g., ACD/ChemSketch, Hyper Chem, E-Dragon and AlogPS 2.1. These computational methods are fast and cheaper because they do not require chemical reagents, apparatus or laboratory study. However, in order to obtain reliable lipophilicity results, the theoretical, i.e., calculated lipophilicity parameters should be compared with those obtained using the experimental methods, e.g., chromatography. It can be seen that the older and more time-consuming shake flask method is often replaced by chromatographic techniques, including reversed-phase thin-layer chromatography (RP-TLC) and reversed-phase high-performance thin-layer chromatography (RP-HPTLC) [1]. The main advantages of this technique are the low cost of experiments, simplicity and high precision, and it enables the analysis of many substances simultaneously compared to other chromatographic methods. Interesting review papers focusing on the latest achievements in the measurement of lipophilicity parameter by both, i.e., chromatographic and theoretical methods were presented by Liang and Lian [22], as well as by Kempińska et al. [23]. An extensive literature review confirms that thin-layer chromatography has been successfully applied in assessing the lipophilicity of different bioactive compounds, such as metformin and phenformin [4], triazole derivatives [5,15], steroid compounds [5,6], betulin-1,4-quinone hybrids [7], betulin derivatives [8], quinoline sulfonamides [9], quinoline derivatives [10], thiosemicarbazides [11], acridine derivatives [12], cephalosporines [13], selected chalcones and flavonoids [14], statins [16], natural styryl lactones [17], phenylethylamine drug analogues [18], acetylenequinoline derivatives [19], purine-2,6-dione derivatives [20] and quaternary (fluoro)quinolones [21]. 

The lipophilicity parameter determined in RP-TLC and RP-HPTLC systems is R_MW_ value calculated by the extrapolation of experimental R_M_ value to the zero concentration of organic modifier in applied mobile phase according to Wachtmeister–Soczewiński’s methodology [1]. 

The main aim of the present study was to evaluate the lipophilicity parameters of different bioactive compounds belonging to two groups, namely to substances lowering the concentration of uric acid in the blood and anti-androgenic drugs used in the treatment of prostate cancer. Some of them are relatively new drug substances, such as ailanthone [24], therefore in the available literature there are no experimental lipophilicity parameters for them. 

### 1.1. Uric Acid Lowering Compounds

The first studied group consisted of xanthine oxidase inhibitors, such as allopurinol, its metabolite–oxypurinol and febuxostat, which are successfully used to treat gout, as shown in Figure 1. As is known, gout is a disease that most often causes arthritis. It is characterized by hyperuricemia, which means an overly high concentration of uric acid in the blood. Hyperuricemia is most often the result of impaired uric acid excretion and is associated with diseases, such as chronic nephritis, cardiovascular disease or diabetes mellitus [25]. Among the drug substances tested, allopurinol and febuxostat are competitive inhibitors of the xanthine oxidase, the enzyme responsible for the production of uric acid [25].

### 1.2. Anti-Androgens

The second group of tested compounds were the selected anti-androgens (Figure 2) used to treat prostate cancer. It is one of the most common cancers observed in men over 50 years old worldwide [26]. 

The discovery of the androgen receptor in the late 1960s enabled the development of anti-androgen drugs [27]. This is a chemically diverse group of drugs that plays a key role in the functioning of prostate cancer cells [28,29]. The general mechanism of anti-androgen drugs action is to compete with endogenous androgens in binding to the androgen receptor. The studied abiraterone, bicalutamide, flutamide, nilutamide and ailanthone belongs to these compounds [27,28,29,30,31,32,33]. This group of drugs is constantly being studied and new anti-androgens are being discovered, which are the subject of numerous studies of effectiveness and safety, such as ailanthone, a compound that is currently not used as a drug but is being studied for effectiveness in the treatment of prostate cancer [30,31]. For example, abiraterone is a potent and irreversible inhibitor of the CYP17 enzyme responsible for androgen synthesis. It was approved by the Food and Drug Administration Agency (FDA) in 2011 and has shown significant improvements in overall survival in patients compared to placebo in studies [32]. Next substance, namely bicalutamide, is a non-steroidal anti-androgen drug belonging to the second generation of these drugs and it was approved in 1995 [28]. In this study, we also examined the drugs leflunomide and its active metabolite teriflunomide, which are being studied as effective drugs in the chemoprevention of prostate cancer [32]. 

In our work, the experimental values of the lipophilicity descriptor in the form of the chromatographic parameters (R_MW_) for all tested compounds belonging to both the pharmacological groups were determined. The applied technique was thin-layer chromatography (RP-TLC/HPTLC) using different types of mobile phases and stationary phases, such as RP18F_254_, RP18WF_254_ and RP2F_254_ plates and a mixture of ethanol–water, propan-2-ol and acetonitrile–water. Experimental data have been compared with calculated logP, obtained using different computer software, e.g., AlogPs, AClogP, AlogP, MlogP, XlogP2, XlogP3 ACD/logP and logP_KOWWIN_. The conducted in silico studies allowed the calculation of the theoretical partition coefficient (logP) and other physicochemical factors important in describing the pharmacokinetic properties of studied drug substances, such as density, boiling point, index of refraction, molar refractivity, polar surface area, polarizability, surface tension and molar volume. To the best of the authors’ knowledge, this work is, for the first time, presenting the lipophilicity parameters, experimentally obtained by using TLC method for: febuxostat and oxypurinol, which belong to agents that lower blood uric acid, and also for abiraterone, ailanthone and teriflunomide from group anti-androgen substances. Chemometric methods, i.e., cluster analysis (CA) and principal component analysis (PCA) of obtained results were successfully applied to compare the tested compounds, taking into account both experimental and theoretical lipophilicity parameters, as well as other physicochemical descriptors. In addition, a relatively novel non-parametric method, such as the sum of ranking differences, was used to rank all applied approaches to determine of lipophilicity parameters 

## 2. Results and Discussion

Considering the importance of lipophilicity parameters in the description of the behavior of drug substances in the biological system, the present study focuses on the determination of experimental and theoretical values of lipophilicity parameters of different bioactive compounds as drug substances belonging to agents that lower blood uric acid (allopurinol, oxypurinol, febuxostat) and anti-androgenic compounds (abiraterone, bicalutamide, flutamide, nilutamide, teriflunomide, leflunomide and ailanthone), respectively. The chemical structures of the tested compounds are shown in Figure 1 and Figure 2. Table 1 and Table 2 present the results of in silico study, i.e., the theoretical partition coefficient values (logP) of studied compounds and average logP values (logP_avg_) calculated by using different computer software (AlogPs, AlogP, AClogP, MlogP, xlogP2, xlogP3) by means of the Virtual Computational Chemistry Laboratory, while logP_KOWWIN_ and ACD/logP values were obtained from the EPIWEB program. In addition, Table 1 presents the experimental values (logP_exp_) available for the selected studied compounds. Table 2 presents other physicochemical descriptors of studied compounds, such as density, boiling point, index of refraction, molar refractivity, polar surface area, polarizability, surface tension and molar volume. Depending on the algorithm used, a certain difference was observed between the theoretical logP values and average values of logP (logP_avg_), especially for allopurinol from the group of drugs, lowering the concentration of uric acid in blood and ailanthone as the member of the second analyzed group, namely anti-androgen agents. Significant differences can be observed between the experimental logP value available for allopurinol (logP_exp_ = −1.80) and the average theoretical logP value (logP_avg_ = 0.17). For other compounds with a known logP_exp_ value, the difference between both logP values is smaller. The greatest agreement between the both parameters is observed for bicalutamide. This situation indicates the predictive power of algorithms used to determine the reliable logP value of this compound.

Table 2 illustrates other physicochemical parameters of the tested compounds calculated using computer programs. Differences between these parameters, such as surface tension and molecular volume in the two groups of studied compounds, can be explained by differences in their chemical structure, which was used to predict these parameters. The aim of further research was the chromatographic analysis of blood uric acid lowering and anti-androgen agents. Various systems, consisting of silica gel plates RP18F_254_, RP18WF_254_ and RP2F_254_, and three mobile phases, containing ethanol, propan-2-ol, acetonitrile and water, in different volume compositions were used to determine the chromatographic parameter of lipophilicity in the form of R_MW_, according to the equation of Wachtmeister–Soczewiński (1). The R_MW_ values for all compounds are listed in Table 3. The characteristics of linear relationships between chromatographic parameters, i.e., R_M_ of each examined compound and the content of organic modifier in the mobile phase (φ) used to determine the R_MW_ values in all chromatographic systems are presented in Appendix A. 

The analysis of chromatographic parameters of lipophilicity presented in Table 3 shows that in the group of the tested substances that lower blood uric acid, the highest lipophilicity parameter shows febuxostat, a compound whose experimental value of lipophilicity descriptor has not been well described so far (Table 1). The obtained results may, therefore, be helpful in assessing the lipophilic property of this drug substance. 

Data summarized in Table 3 show the R_MW_ values for febuxostat in the range of 1.9788–2.7106 (ethanol–water), 1.2259–1.9112 (propan-2-ol-water) and 1.5489–1.9061 (acetonitrile–water). For comparison, significantly lower values of chromatographic lipophilicity parameters for febuxostat obtained by the same RP-TLC systems were observed for the two other compounds belonging to the same pharmacological class, i.e., for allopurinol and its metabolite oxypurinol. The chromatographic results of R_MW_ values are the next group of compounds depicted in Table 3, close to their theoretical logP values (Table 1), thus indicating the similarity between the R_MW_ and logP values of all anti-androgens, except for abiraterone and ailanthone, i.e., between bicalutamide, flutamide, nilutamide, leflunomide and teriflunomide. The first of them, i.e., abiraterone shows the highest R_MW_ value in the range of 1.8373–4.5660. Of all the R_MW_ results, those obtained with ethanol–water and the three chromatographic plates ranging from 3.8765–4.5660 best correspond to the theoretical logP values predicted using different algorithms, as well as to the average value of them. In the case of ailanthone, all obtained R_MW_ results are lower compared to other anti-androgens, which is also observed between logP values. The greatest similarity can be observed between the chromatographic parameter of lipophilicity (R_MW_ = 0.9749) measured by RP-TLC method on RP18F_254_ plates and acetonitrile–water as the mobile phase and the theoretical logP value marked as MlogP = 0.99. A similar agreement indicates R_MW_ = 0.2488 achieved on RP18W_254_ plates developed by propan-2-ol and logP_KOWWIN_ (0.25). The third examined anti-androgen whose logP_exp_ is not available, i.e., teriflunomide, shows the best similarity of the R_MW_ values obtained by propan-2-ol-water (2.0438) to AlogP value (2.07), as well as with other theoretical partition coefficients, such as AlogPs (2.30) and logP_KOWWIN_ (2.25). This fact confirms the usefulness of these partition coefficients to evaluate the lipophilicity of the presented compound. 

The analysis of the R_MW_ values of bicalutamide, nilutamide, flutamide and leflunomide presented in Table 3 confirms the usefulness of almost-tested chromatographic systems for the evaluation of the lipophilicity of these compounds. However, the best are those consisting of ethanol–water and acetonitrile–water, as well as the chromatographic plates RP18F_254_ and RP18WF_254_. These R_MW_ results are very similar to logP_exp_ and to the average value of theoretical logP given in Table 1. In the further parts of our study, we used the two chemometric methods, such as cluster analysis and principal component analysis, to compare the studied compounds belonging to different pharmacological classes on the basis of all the parameters presented in this work, including the experimental, i.e., chromatographic parameters of lipophilicity, theoretical logP values calculated using different algorithms, as well as other physicochemical descriptors predicted on the basis of their structure, i.e., density, boiling point, index of refraction, molar refractivity, polar surface area, polarizability, surface tension and molar volume. All data are presented in Table 1 and Table 2, respectively. These data were used in the similarity analysis of analyzed compounds. The first analysis concerned the similarity of these compounds in terms of lipophilicity values obtained experimentally, as shown in Figure 3A, and theoretical values, as shown in Figure 3B. 

Next, Figure 3 presents the similarity analysis of tested compounds based on their experimentally obtained lipophilicity parameters (R_MW_) and theoretical logP. The analysis of both dendrograms indicates that three clusters can be distinguished in Figure 3A, consisting of oxypurinol and allopurinol; teriflunomide and febuxostat; and nilutamide, leflunomide, bicalutamide and abiraterone, respectively. On the other hand, the analysis of only theoretical logP values (Figure 3B) allows the separation of two clusters, including, respectively, oxypurinol, allopurinol and ailanthone, as well as other compounds, except for abiraterone. As well as basing on these analyses, it is possible to determine the differences between the experimental and theoretical values of lipophilicity. The theoretical ones only take into account the structure of the compounds, hence, for example, the very high similarity of compounds from a larger cluster, where all (except febuxostat) have fluorine atoms in their composition. Considering the Euclidean distance, these two groups of compounds (two clusters) are also far apart, while the experimental values are much more similar to each other. In Figure 3A, three visible clusters can be seen, which include, respectively, oxypurinol and allopurinol; febuxostat and teriflunomide; and nilutamide, leflunomide, flutamide and bicalutamide. The first cluster includes compounds that are similar both in terms of structure and action because both belong to the group of drugs used in gout. The second cluster includes compounds containing a carbon–nitrogen triple bond in the structure, which may cause similar interactions between the compound and the mobile or stationary phase during TLC analysis used to determine logP. When data on the values of other physicochemical properties were added to the cluster analysis, the analysis allowed us to distinguish three clusters, although the Euclidean distances, in this case, are much larger than in the case of the logP similarity analysis alone (Figure 4). 

The resulting clusters, including bicalutamide and ailanthone; flutamide and teriflunomide; and nilutamide and febuxostat, were based on the similarity of their physicochemical properties and they have the final influence on the resulting graph. To describe the variability of the tested system, an analysis of principal components was also carried out, during which all data obtained for the tested compounds were taken into account, namely theoretical and experimental values of lipophilicity and values of physicochemical properties. All data have been standardized. First, the eigenvalues were extracted (Table 1). Based on the Kaiser criterion, it was found that the first three eigenvalues are sufficient to describe the examined set of compounds because their value exceeds 1. They describe 92.68% of the variability of the system. 

Using the principal components, a graph of the projection of cases onto the plane of factors was prepared. The graph is shown in Figure 5. 

The clusters marked in Figure 6 confirm what the cluster analysis showed for all the analyzed data. These clusters contain the same compounds as the clusters in Figure 5. This indicates the high usefulness of the PCA analysis, which allows only three principal components to be taken into account instead of many data describing the system, and the conclusions of the analysis remain the same. Similar conclusions can also be observed when analyzing all logP data, experimental and theoretical. PCA for these values would reduce the amount of data to two eigenvalues. On the basis of these two eigenvalues, it is possible to draw a graph of the projection of cases onto the plane of factors, which would confirm, as in the previous case, the similarity analysis for these data. The figure below shows the graphs prepared based on PCA (Figure 6A) and based on CA (Figure 6B). 

Continuing this study, we used novel non-parametric methods, such as ranking (SRD analysis), to obtain a reliable comparison of lipophilicity parameters of examined compounds. The inspiration of this analysis was the recently published papers on the determination of lipophilicity of various bioactive compounds by chromatographic and theoretical methods, such as acetylenequinoline derivatives [19], purine-2,6-dione based compounds [20] and quaternary (fluoro)quinolones [21]. However, it should be highlighted that the best guide regarding the procedure for conducting the SRD analysis are the pieces of work published by Héberger and Andrić [34,35,36]. The SRD ranking of several chromatographically estimated lipophilicity descriptors expressed as R_MW_ values and computationally calculated logP values of all tested compounds shown in Figure 7, confirming the best lipophilicity measures and the closest to the zero shows the theoretical partition coefficient MlogP. Thus, it can be concluded that the theoretical parameter of lipophilicity MlogP may be considered a suitable lipophilicity measure for studied compounds, whereas the calculated XlogP2 and XlogP3 variables have the highest SRD scores of all theoretically obtained logP values. It can be also seen that the chromatographic parameter of lipophilicity denoted as RP18W(EtOH/H_2_O), which was determined using ethanol–water on chromatographic plates precoated with silica gel RP18WF_254_, falls in the same range as MlogP, i.e., near zero. According to the SRD values, ethanol would be suggested as the most suitable organic modifier for the determination of chromatographic parameters of lipophilicity, i.e., the R_MW_ of studied compounds. 

## 3. Materials and Methods

### 3.1. Reagents

Ethanol (96%, Reag. Ph Eur.), dimethyl sulfoxide (DMSO), propan-2-ol and acetonitrile of HPLC grades were bought from POCh (Gliwice, Poland). Deionized water was produced using the Direct-Q3 UV system (Millipore, Warsaw, Poland). 

### 3.2. Analytes

The reference standards of anti-androgen compounds (purity ≥ 98%), such as abiraterone acetate, bicalutamide, flutamide, nilutamide, teriflunomide, leflunomide and ailanthone, were purchased from Sigma-Aldrich (Beijing, China). The standards (purity ≥ 98%) of blood urea lowering agents, i.e., allopurinol, oxypurinol and febuxostat, were obtained from Sigma-Aldrich (Rehovot, Israel). The studied compounds were dissolved in DMSO or in ethanol, respectively, to obtain a concentration of 5 mg/mL. The stock solutions were stored at 2–8 °C prior to analyses. 

### 3.3. Materials

Chromatographic analysis was performed on aluminum plates (20 cm × 20 cm) precoated with silica gel RP18F_254_ and glass plates coated with silica gel RP18WF_254_ (20 cm × 10 cm) and silica gel RP2F_254_ (10 cm × 10 cm), manufactured by Merck (Darmstadt, Germany). 

### 3.4. Chromatographic Analysis

Chromatographic analysis was conducted on RP-TLC plates (RP2F_254_, RP18F_254_, RP18WF_254_). Five microliters of standard solutions of analytes were spotted onto the chromatographic plates. The chromatographic chamber of 20 cm × 10 cm (Camag, Muttenz, Switzerland) was saturated with the mobile phase vapors for 20 min. The mobile phases were prepared by mixing appropriate organic modifiers (ethanol, propan-2-ol, acetonitrile) and water in different volume compositions in a range from 20% to 90% (*v*/*v*). The content of the organic modifier in the mobile phase used was changed every 5% (*v*/*v*). Chromatograms were developed at room temperature (21 ± 1 °C) to the solvent distance of 7 cm. Next, the chromatograms were dried for 24 h at 21(±1 °C) in a fume cupboard. The identification of studied compounds were carried out under a UV lamp at 254 nm (Camag, Switzerland). The values of R_f_ (retardation factor) are the average values of three independent measurements in each case. To determine the chromatographic parameter of lipophilicity of studied compounds in terms of R_MW_ value, the Soczewiński-Wachtmeister’s [1] equation, which shows the linear relationship between the chromatographic factor R_M_ and volume fraction of organic modifier in the mobile phase (φ), was used [1]: (1)$$R_{M}=R_{MW}-b\times \varphi$$

### 3.5. In Silico Study

Various software packages with different calculation algorithms were useful for the prediction of the logP values, as well as other physicochemical parameters of studied compounds. The source of the following physicochemical descriptors of studied compounds, such as density, boiling point, index of refraction, molar refractivity, polar surface area, polarizability, surface tension and molar volume, given in Table 2, were EPIWEB 4.1 program (Estimation Programs Interface) Suite TM Version 4.1 and ChemSpider (http://www.chemspider.com), accessed on 19 October 2022. Six different logP values (AlogPs, AlogP, AClogP, MlogP, xlogP2, xlogP3) and average logP value (logP_avg_) were predicted by using Virtual Computational Chemistry Laboratory http://www.vcclab.org./ (accessed on 20 October 2022), while logP_KOWWIN_ and ACD/logP values were obtained from ChemSpider (http://www.chemspider.com), accessed on 20 October 2022. In addition, the experimental value (logP_exp_) available for selected studied compounds, such as allopurinol, bicalutamide, flutamide, nilutamide and leflunomide, are presented in this work (Table 1). The logP_exp_ values were derived from a drug database online, namely DrugBank (https://www.drugbank.com/), accessed on 20 October 2022. All the experimental and calculated logP values were summarized in Table 1. 

### 3.6. Cluster Analysis (CA)

Cluster analysis, in the simplest terms, allows you to group objects that are similar to each other [37], an accessible course in statistics with the use of Statistica PL. Thanks to such grouping of objects (data), we found that the data in one cluster or group (cluster) indicate some regularity. Cluster analysis in the presented work was performed using Statistica 13.3 software. During the analysis, the calculations were based on Euclidean distances and the single linkage distance. 

### 3.7. Principal Component Analysis (PCA)

Principal component analysis is useful when a given system is described by many variables. Through PCA, the number of variables can be reduced to the minimum number necessary to describe the variability of the system [38]. The analysis of principal components in the presented work was carried out using the Statistica 13.3 software. The number of eigenvalues was determined based on the Kaiser criterion and scree plot. All data for principal component analysis were previously standardized. The purpose of such standardization is to change the value of data in such a way that it will allow for their analysis when they have different dimensions. 

### 3.8. Sum of Ranking Differences (SRD)

Sum of ranking difference is a non-parametric method useful for performing a reliable comparison of methods for the measurements or calculations of the same property, such as lipophilicity descriptors [34,35,36]. The SRD analysis of both chromatographically obtained values using various TLC systems (different stationary and mobile phases) and calculated lipophilicity parameter values of examined compounds was performed using Microsoft Excel macro program downloaded at http://aki.ttk.mta.hu/srd/ (accessed on 15 December 2022). 

## 4. Conclusions

The chromatographic studies presented in this paper emphasize the importance of thin-layer chromatography in the reversed-phase system for the evaluation of the lipophilicity of selected compounds that lower blood uric acid and anti-androgen compounds, including those for which there is no experimental value of the partition coefficient as a measure of their lipophilicity. The following stationary phases, such as RP18F_254_, RP18WF_254_ and RP2F_254_ and three mobile phases consisting of ethanol–water, propan-2-ol-water, as well as acetonitrile–water, as mobile phases, were successfully applied in this work. The chromatographic parameters of lipophilicity were compared with logP values obtained with different software packages, such as AClogP, AloGPs, AlogP, MlogP, XlogP2, XlogP3, ACD/logP and logP_KOWWIN_.

The similarity between the R_MW_ values obtained onto three chromatographic plates with ethanol–water and acetonitrile–water as mobile phases with logP_exp_ or with average values of theoretical logP confirms the usefulness of the proposed chromatographic method for measuring this important parameter in the point of view of pharmacokinetic properties of drug substances, such as lipophilicity. The results obtained also show that chemometric methods, such as PCA and CA, can be a good tool to illustrate the similarity and differences existing between the investigated compounds based on their both, i.e., experimentally and theoretically obtained lipophilicity parameters, as well as other physicochemical descriptors, including, e.g., density, boiling point, index of refraction, molar refractivity, polar surface area, polarizability, surface tension and molar volume. It has been found that the SRD analysis is very useful in obtaining the reliable comparison of chromatographically determined and theoretical parameters of all studied compounds belonging to compounds that lower blood uric acid and anti-androgenic compounds. The SRD analysis confirms that of all calculation methods, the method based on the MlogP algorithm is the most useful for determining the best lipophilicity measures of the tested compounds. According to SRD scores, ethanol would be suggested as the most suitable organic modifier for the determination of chromatographic parameters of lipophilicity, i.e., R_MW_ of studied compounds using thin-layer chromatographic method in a reversed-phase system.

The lipophilicity parameters obtained in this work can be successfully applied in further study, i.e., in quantitative structure–property relationship and quantitative structure–activity relationship, as well as to facilitate the process of designing new derivatives of the examined compounds that lower blood uric acid or anti-androgens, respectively.

## Figures and Tables

**Figure 1 molecules-28-00166-f001:**
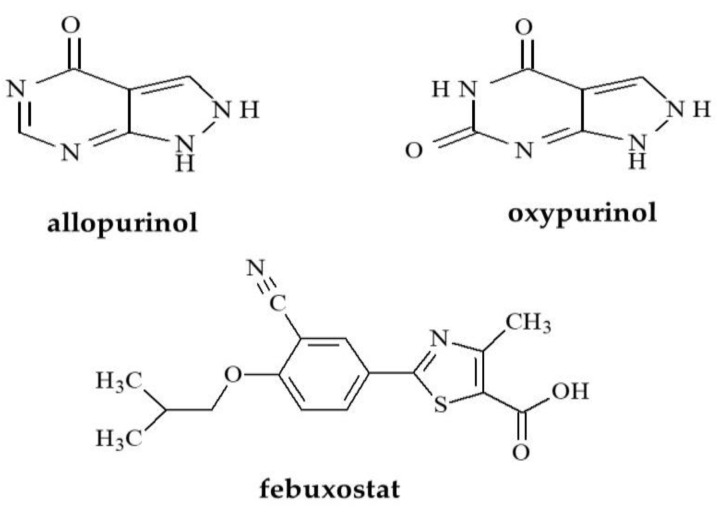
Chemical structures of studied compounds that lower blood uric acid.

**Figure 2 molecules-28-00166-f002:**
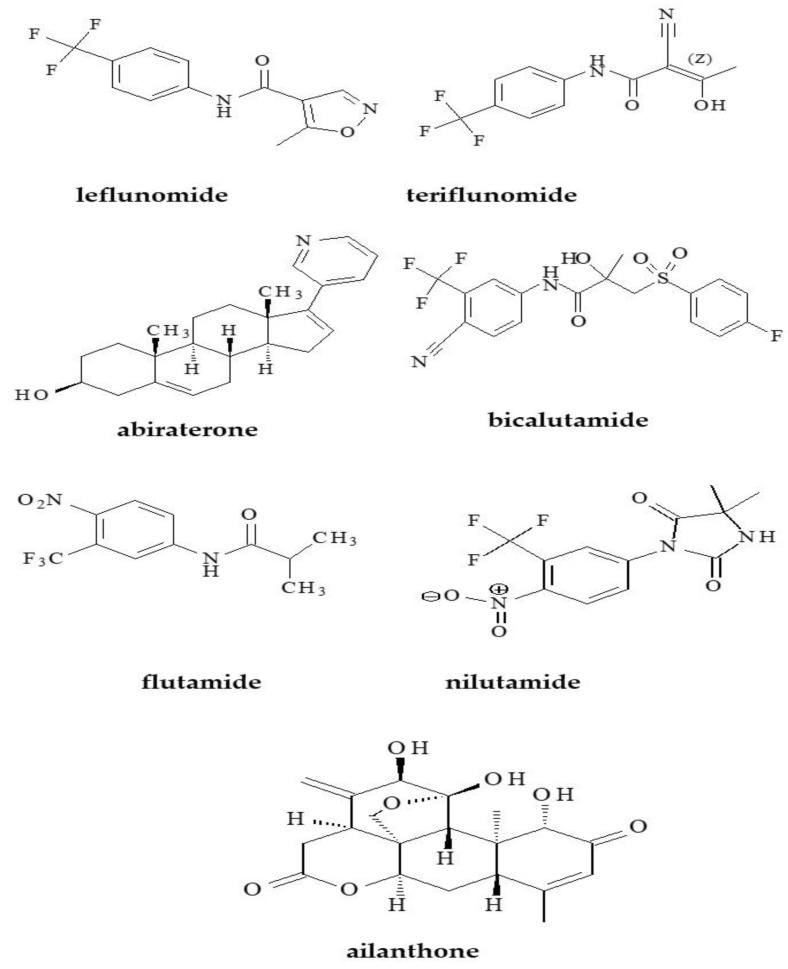
Chemical structures of studied anti-androgen substances.

**Figure 3 molecules-28-00166-f003:**
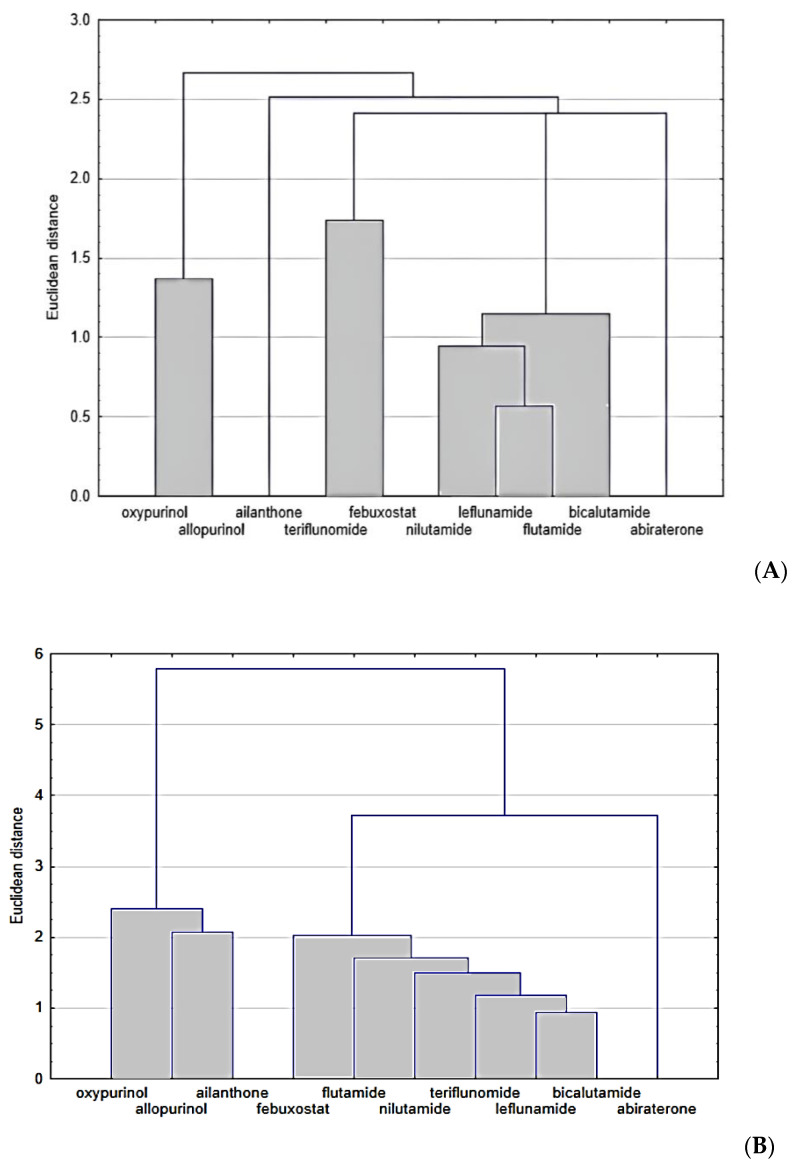
Similarity analysis of tested compounds based on their experimentally determined parameters of lipophilicity (**A**) and theoretical logP (**B**).

**Figure 4 molecules-28-00166-f004:**
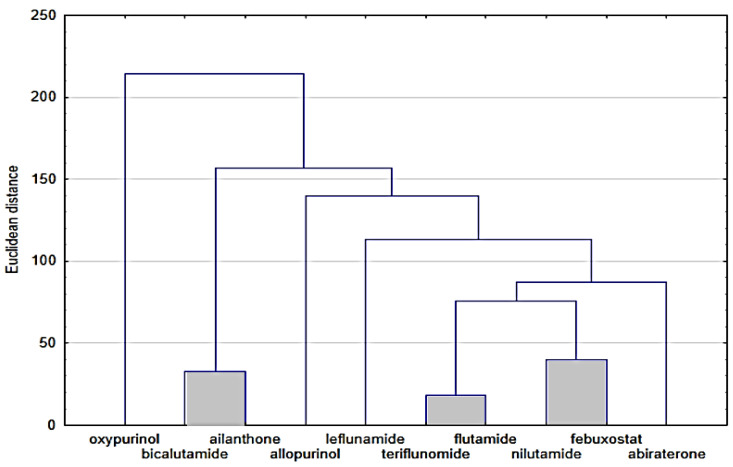
Analysis of similarities of studied compounds based on the analysis of all data regarding both lipophilicity and physicochemical properties.

**Figure 5 molecules-28-00166-f005:**
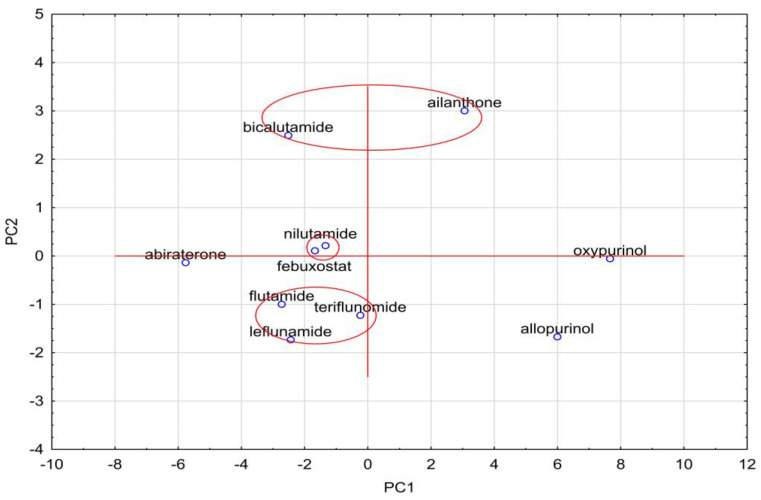
Plot of case projections on the plane of factors for all analyzed data.

**Figure 6 molecules-28-00166-f006:**
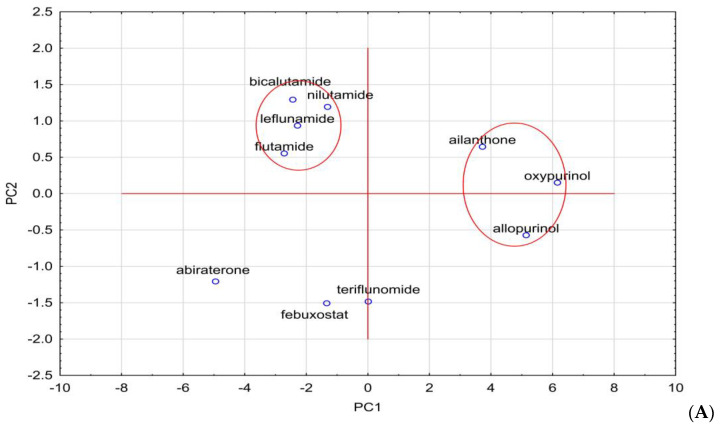
Graph prepared on the basis of principal component analysis (**A**) and cluster analysis (**B**) for all tested compounds in terms of their lipophilicity values, both experimental and theoretical.

**Figure 7 molecules-28-00166-f007:**
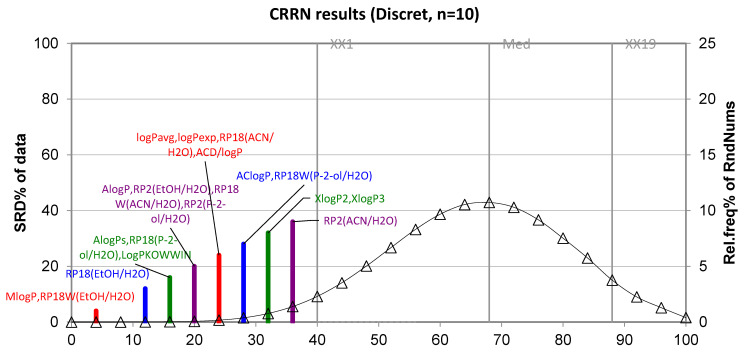
SRD analysis of both; chromatographically obtained and theoretical values of lipophilicity descriptors of studied compounds.

**Table 1 molecules-28-00166-t001:** Partition coefficients of studied compounds.

Compound	AlogPs	AC Logp	AlogP	MlogP	XlogP2	XlogP3	logP_avg_	logP_exp_	logP _KOWWIN_	ACD/logP
** *Blood Uric Acid Lowering Compounds* **										
**Allopurinol**	−0.41	−0.20	0.03	0.58	−0.90	0.15	0.17	−1.80	−1.14	−1.46
**Oxypurinol**	−0.65	−0.44	−0.87	−0.73	−0.34	−0.93	−0.66	-	−2.17	−1.35
**Febuxostat**	3.80	4.09	3.44	2.00	2.60	3.90	3.31	-	4.21	-
** *Anti-Androgens* **										
**Abiraterone**	5.10	4.17	4.22	4.52	4.02	4.63	4.44	-	6.40	5.70
**Bicalutamide**	2.70	2.00	2.93	2.74	2.53	2.31	2.53	2.50	2.30	4.94
**Flutamide**	2.55	3.02	2.92	3.16	2.64	3.35	2.94	3.35	3.51	3.72
**Nilutamide**	1.74	2.08	2.26	2.23	1.84	2.00	2.02	1.80	1.92	2.23
**Leflunomide**	2.52	2.45	2.16	2.37	2.79	2.49	2.46	2.80	2.43	-
**Teriflunomide**	2.30	3.06	2.07	1.68	-	3.27	2.48	-	2.25	2.51
**Ailanthone**	0.01	0.26	−0.32	0.99	−0.47	−1.12	−0.11	-	0.25	−0.76

**Table 2 molecules-28-00166-t002:** Different physicochemical parameters of tested compounds.

Compound	Density [g/cm^3^]	Boiling Point [°C]	Index of Refraction	Molar Refractivity	Polar Surface Area [A°]	Polarizability [cm^3^]	Surface Tension [dyne/cm]	Molar Volume [cm^3^]
** *Blood Uric Acid Lowering Compounds* **								
**Allopurinol**	1.7	423.2	1.816	34.7	75	13.8	126.4	80.0
**Oxypurinol**	1.9	662.9	1.903	36.6	95	14.5	170.2	78.4
**Febuxostat**	1.3	488.2	1.605	83.1	83	32.9	63.7	240.9
** *Anti-Androgens* **								
**Abiraterone**	1.1	500.2	1.606	105.2	33	91.7	50.1	305.2
**Bicalutamide**	1.5	650.3	1.578	93.8	116	37.2	58.2	282.8
**Flutamide**	1.4	400.3	1.521	61.3	75	24.3	38.3	201.3
**Nilutamide**	1.5	477.3	1.524	66.3	95	26.3	42.9	216.8
**Leflunomide**	1.4	289.3	1.541	61.0	55	24.2	40.6	194.1
**Teriflunomide**	1.4	410.8	1.552	60.6	73	24.0	45.4	189.7
**Ailanthone**	1.0	641.0	1.640	91.9	113	36.4	68.0	254.9

**Table 3 molecules-28-00166-t003:** Comparison of R_MW_ values of studied compounds obtained with the use of different stationary phases in the RP-TLC/RP-HPTLC systems.

Mobile Phase	Chromatographic Plates		
	RP2F_254_	RP18F_254_	RP18WF_254_
** *Blood Uric Acid Lowering Compounds* **			
**Allopurinol**			
Ethanol–water	0.4000	0.2164	0.2321
Propan-2-ol-water	−0.0849	0.0399	−0.0722
Acetonitrile–water	0.2783	−0.0737	−0.1744
**Oxypurinol**			
Ethanol–water	−0.5431	0.1808	−0.0346
Propan-2-ol-water	−0.1945	−0.3749	−0.5789
Acetonitrile–water	0.7764	0.3765	−0.2654
**Febuxostat**			
Ethanol–water	1.9788	2.7106	2.0687
Propan-2-ol-water	1.2524	1.9112	1.2259
Acetonitrile–water	1.5648	1.9061	1.5489
** *Anti-Androgens* **			
**Abiraterone**			
Ethanol–water	4.3012	4.5660	3.8765
Propan-2-ol-water	2.5179	3.0060	2.0141
Acetonitrile–water	2.7366	1.8373	2.4400
**Bicalutamide**			
Ethanol–water	2.8044	3.1861	2.8711
Propan-2-ol-water	1.9488	2.7550	1.4439
Acetonitrile–water	3.0843	4.0113	2.8946
**Flutamide**			
Ethanol–water	2.9756	3.1037	2.9650
Propan-2-ol-water	2.4137	2.2432	1.7854
Acetonitrile–water	2.6436	3.3395	2.7053
**Nilutamide**			
Ethanol–water	3.0454	2.8873	2.3245
Propan-2-ol-water	2.1987	1.9313	1.4803
Acetonitrile–water	2.7675	3.1543	2.3260
**Leflunomide**			
Ethanol–water	3.0335	3.5015	2.8223
Propan-2-ol-water	2.5391	2.3386	1.8840
Acetonitrile–water	2.7559	3.0430	2.6675
**Teriflunomide**			
Ethanol–water	1.3228	1.9800	1.2736
Propan-2-ol-water	1.0884	2.0438	1.8615
Acetonitrile–water	0.9696	1.3186	1.0250
**Ailanthone**			
Ethanol–water	0.8299	0.6137	0.3212
Propan-2-ol-water	0.7606	1.7092	0.2488
Acetonitrile–water	0.4203	0.9749	1.2568

## Data Availability

Not applicable.

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
