# Peer review of "Comparative Study of the Lipophilicity of Selected Anti-Androgenic and Blood Uric Acid Lowering Compounds"

_molecules, 2022, doi:10.3390/molecules28010166_

Round 1

Reviewer 1 Report

The work is very organized and interesting. However, minor recommendation should be performed before final publication

1-     Reorganization of the introduction is strongly recommended into separate paragraphs  

2-     At line 70, the drug groups studied were chosen because ….;

 Start new paragraph from this point to highlight the reasons for choosing such drugs

3-     Some important references were missed , please add them

-        https://doi.org/10.1016/j.trac.2019.01.011

-        State of the art and prospects of methods for determination of lipophilicity of chemical compounds

-        https://doi.org/10.1016/j.trac.2015.02.009

-        Recent advances in lipophilicity measurement by reversed-phase high-performance liquid chromatography

-        Any related articles could be added for increasing number of references.

4-     Future research prospective and application of the outcomes for this study should be added after discussion

5-     Statement about the most reliable tools for prediction of drug lipophilicity should be highlighted in the conclusion

Best wishes

Author Response

Review Report Form 1

Dear Reviewer,

The authors are very grateful for manuscript revision and all pertinent comments which improved the quality of manuscript. We made the changes and improved the English language. All changes are highlighted in yellow. We hope that all corrections will meet with your requirements.

Below we present the authors responses to questions.

Comments and Suggestions for Authors

The work is very organized and interesting. However, minor recommendation should be performed before final publication

Comment No1.  and No. 2

1-     Reorganization of the introduction is strongly recommended into separate paragraphs  

2-     At line 70, the drug groups studied were chosen because ….;

 Start new paragraph from this point to highlight the reasons for choosing such drugs

Answer to Comments No 1  and 2

Dear Reviewer, we are very grateful for this important suggestion. Based on the both reviews, which were similar in this point, we have reorganized manuscript in Introduction part.  Introduction part was devided into to paragraphs. Chemical structures of studied compounds have been also introduced in accordance to review 2.  We have clearly highlighted the importance of studied compounds in this section  i.e. we highlighted that some of them are new drug candidates, especially the ailanthone. Therefore, it a lack of their experimental parameter of lipophilicity.

We hope that it will meet with your requirements.

Comment No 3.

3-     Some important references were missed , please add them

-        https://doi.org/10.1016/j.trac.2019.01.011

-        State of the art and prospects of methods for determination of lipophilicity of chemical compounds

-        https://doi.org/10.1016/j.trac.2015.02.009

-        Recent advances in lipophilicity measurement by reversed-phase high-performance liquid chromatography

-        Any related articles could be added for increasing number of references.

Answer to Comment No 3 

Dear Reviewer, thank you very much for this remark. We made extensive literature review again and the two highlighed review papers as well as additional published during the last years were implemented to current manuscript as Ref [22] and Ref. [23] in References list and in text – page 2. Thanks to this suggestion, increased the number or cited paper to 38.

Comment No 4.                                                                                                  

  1. Future research prospective and application of the outcomes for this study should be added after discussion

Answer to Comment No 4 

Dear Reviewer, thank you very much for this remark. Future research prospective and application of the outcomes for this study  was added in Conclusion part (see page 15).

Comment No 5.

5-     Statement about the most reliable tools for prediction of drug lipophilicity should be highlighted in the conclusion

Answer to Comment No 5 

Dear Reviewer we are grateful for this comment. In Conclusion part  we added these informations.

Reviewer 2 Report

The paper entitled “Comparative Study of the Lipophilicity of Selected Anti-Androgenic and Blood Uric Acid Lowering Compounds” authored by Dawid Wardecki, MaÅ‚gorzata DoÅ‚owy, Katarzyna Bober-Majnusz and Josef Jampilek presents the evaluation of the lipophilicity of a series of blood uric acid lowering compounds and anti-androgen substances applying thin-layer chromatography in reversed-phase system (RP-TLC, RP-HPTLC) and computational methods. The authors applied three stationary phases, three types of mobile phase. The chromatographic analysis is conducted properly and the results are suitably evaluated. The paper is well written and nicely organized, however some changes should be made.

1. It would be better if the molecular structures are provided in the paper since it is easier for the readers to visually gain an overview of the relationship between the molecular structure and determined lipophilicity;

2. Since the PCA was applied, the word “factor” on the plots of case projections should be changed to PC.

3. Figure 3 should be removed, it is not so crucial for the manuscript;

4. The markings of the clusters on the dendrograms should be changed. Instead of using an ellipse, the clusters or subclusters should be shaded or marked in a different way;

5. The HCA and PCA are basic pattern recognition methods. Since the authors presented quite interesting chromatographic data, I think that additional method should be used in order to compare and rank the experimental and in silico data. I suggest sum of ranking differences analysis (SRD) to be conducted on the obtained data in order to rank and compare the experimentally obtained and computationally calculated lipophilicity data.

Author Response

Review Report 2

Dear Reviewer, the authors are very grateful for all partinent comments to manuscript. They improved the quality of our paper. Corrections we have highlighted in yellow colour. English language was checked and improved. We hope that all corrections made by authors  will meet with your requirements. Below we present the answers for all  questions.

Comments and Suggestions for Authors

The paper entitled “Comparative Study of the Lipophilicity of Selected Anti-Androgenic and Blood Uric Acid Lowering Compounds” authored by Dawid Wardecki, MaÅ‚gorzata DoÅ‚owy, Katarzyna Bober-Majnusz and Josef Jampilek presents the evaluation of the lipophilicity of a series of blood uric acid lowering compounds and anti-androgen substances applying thin-layer chromatography in reversed-phase system (RP-TLC, RP-HPTLC) and computational methods. The authors applied three stationary phases, three types of mobile phase. The chromatographic analysis is conducted properly and the results are suitably evaluated. The paper is well written and nicely organized, however some changes should be made.

Comment No 1.

  1. It would be better if the molecular structures are provided in the paper since it is easier for the readers to visually gain an overview of the relationship between the molecular structure and determined lipophilicity;

Answer to Comment No 1

Dear Reviewer thank you very much for this imporatant remark. The chemical structures of all studied compounds we added to Introduction part (page No. 3 and 4)

Comment No 2.

  1. Since the PCA was applied, the word “factor” on the plots of case projections should be changed to PC.

Answer to Comment No 2

Dear Reviewer we are very grateful for this important attention. We made correction of Figures (page No. 11)

Comment No 3.

  1. Figure 3 should be removed, it is not so crucial for the manuscript;

Answer to Comment No 3.

Dear Reviewer thank you for this remark. We removed this Figure in current manuscript.

Comment No 4.

  1. The markings of the clusters on the dendrograms should be changed. Instead of using an ellipse, the clusters or subclusters should be shaded or marked in a different way;

Answer to Comment No 4.

Dear Reviewer, we are grateful for this suggestion. We made the changes these Figures 3,4 and 6. 

Comment No 5.

  1. The HCA and PCA are basic pattern recognition methods. Since the authors presented quite interesting chromatographic data, I think that additional method should be used in order to compare and rank the experimental and in silico data. I suggest sum of ranking differences analysis (SRD) to be conducted on the obtained data in order to rank and compare the experimentally obtained and computationally calculated lipophilicity data.

Answer to Comment No 5.

Dear Reviewer we are very grateful for this advice concerning the need of implementation to our study of SRD analysis. We made a proper correction (page No. 12,13) thanks to valuable papers published by Heberger and Andricz Ref. 34-36. In addtion to this we learned a new chemometric tool important for this and next our papers.  

Round 2

Reviewer 2 Report

The paper is properly corrected and should be accepted for publication.